# OceanGym: A Benchmark Environment for Underwater Embodied Agents

## Abstract

We introduce OceanGym, the first comprehensive benchmark for ocean underwater embodied agents, designed to advance AI in one of the most demanding real-world environments. Unlike terrestrial or aerial domains, underwater settings present extreme perceptual and decision-making challenges, including low visibility, dynamic ocean currents, making effective agent deployment exceptionally difficult. OceanGym encompasses eight realistic task domains and a unified agent framework driven by Multi-modal Large Language Models (MLLMs), which integrates perception, memory, and sequential decision-making. Agents are required to comprehend optical and sonar data, autonomously explore complex environments, and accomplish long-horizon objectives under these harsh conditions. Extensive experiments reveal substantial gaps between state-of-the-art MLLM-driven agents and human experts, highlighting the persistent difficulty of perception, planning, and adaptability in ocean underwater environments. By providing a high-fidelity, rigorously designed platform, OceanGym establishes a testbed for developing robust embodied AI and transferring these capabilities to real-world autonomous ocean underwater vehicles, marking a decisive step toward intelligent agents capable of operating in one of Earth's last unexplored frontiers.

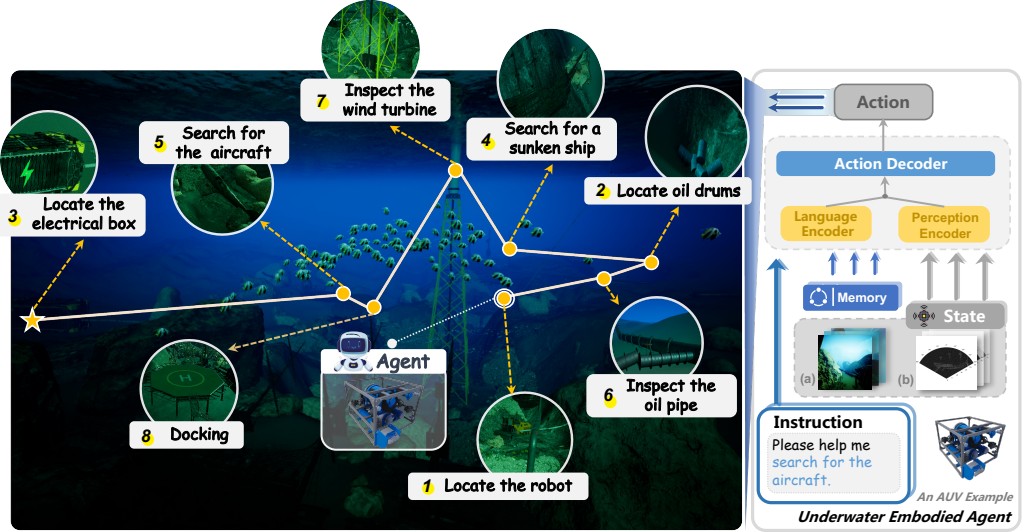

Figure 1: **Illustration of OceanGym.** The OceanGym benchmark introduces a unified **agent framework** across **8 real-world underwater scenarios**. The agent interprets language instruction, fuses optical and sonar imagery, and controls Autonomous Underwater Vehicles (AUVs).

## 1 Introduction

As Richard S. Sutton famously noted, we are entering an "era of experience" (Silver & Sutton, 2025). Embodied agents equipped with language models (Zhao et al., 2023; OpenAI, 2024) are emerging as a central paradigm for intelligent systems, as they accumulate and leverage experience through continuous interaction to close the perception-decision-action loop in physical or simulated environments (Gupta et al., 2021; Ding et al., 2024; Liu et al., 2025). Unlike static decision or generative

models, these agents must integrate rich multimodal sensory streams and execute continuous-control policies to achieve long-horizon objectives. This necessitates a unified treatment of perceptual representation, planning, online inference, and sequential policy optimization (Fung et al., 2025). Significant progress has been demonstrated across diverse domains, including robotic manipulators (Anderson et al., 2018; Caesar et al., 2020; Vasudevan et al., 2021; Gao et al., 2024), drones (Wang et al., 2024a; Lee et al., 2025; Gao et al., 2025b), and autonomous vehicles (Ma et al., 2025b).

In contrast, *underwater*[1] embodied agents remain largely unexplored despite their critical scientific and engineering importance (Visbeck, 2018; Kelly et al., 2022; Zheng et al., 2023; Li et al., 2024b; Gao et al., 2025a). Deploying embodied agents in marine environments offers unique opportunities for ocean exploration, offshore resource development, environmental monitoring, and subsea rescue operations. These tasks impose stringent requirements on the robustness and reliability of autonomous platforms, making the development of agents capable of functioning under real marine conditions a key bridge between simulated research and practical deployment (Ma et al., 2025a).

**Challenges.** Underwater embodied agents face distinct challenges that set them apart from overland and aerial systems. *Perceptually*, poor visibility and low-light conditions, combined with the inherent limitations of optical sensors, compel reliance on sonar, inertial measurements, and other sparse modalities (Li et al., 2024c; Aubard et al., 2025). These heterogeneous and noisy observations complicate sensor fusion and perception. *Environmentally*, deep-sea and offshore settings are largely unexplored, with unstable localization, absent prior knowledge, and dynamic currents. The lack of prior knowledge prevents effective environmental modeling, requiring agents to reason under circumstances of extreme partial observability and uncertainty (Sariman et al., 2025). Together, these factors constrain the development of underwater agents, leaving their capabilities in early stages.

**Building OceanGym.** To address these challenges, we introduce OCEANGYM, an open environment benchmark for underwater embodied agents. OCEANGYM constructs a comprehensive marine environment spanning approximately 800m × 800m (length × width), with dynamically adjustable depth to simulate varying lighting conditions. The platform incorporates eight distinct task domains designed to reflect real-world operational requirements. Additionally, it provides a multimodal LLM-based agent framework that integrates perception, memory, and action decision-making capabilities for controlling Autonomous Underwater Vehicles (AUVs). The benchmark unifies perception and decision-making in simulated underwater scenarios, where agents must infer target states from contextual cues or multi-view sensor data and execute complex behaviors such as search, inspection, and docking. By simulating these environments, OCEANGYM enables systematic evaluation of language models' capabilities in underwater embodied settings and offers a pathway for transferring learned skills to real-world underwater vehicles through the generation of synthetic data, reinforcement learning guided by environmental feedback, and iterative improvement of agent capabilities through various algorithmic approaches. We discuss the limitations of OCEANGYM in §3.3.

**Benchmark Results and Insights.** Extensive experiments on OCEANGYM reveal that Multi-modal Large Language Models (MLLMs) exhibit significant gaps compared to human experts, particularly under low-visibility conditions (decision-making success rate drops to 14.8%). Agents frequently struggle to interpret sonar data accurately, distinguish objects in complex environments, and maintain consistent devision strategies over extended missions. Limitations also arise in memory retention and adaptability when objects are occluded or conditions change dynamically. These findings highlight persistent challenges for embodied AI in underwater environments and underscore the need for continued research in robust perception, reasoning, and control under extreme uncertainty.

## 2 OCEANGYM

OCEANGYM is a high-fidelity embodied underwater environment that simulates a realistic ocean setting with diverse scenes. As illustrated in Figure 2, OCEANGYM establishes a robust benchmark for evaluating autonomous agents through a series of challenging tasks, encompassing various perception analyses and decision-making navigation. OCEANGYM facilitates these evaluations by enabling MLLM-driven agents with multi-modal perception and parameterized action spaces.

---

[1]Underwater refers to the ocean environment throughout this work and is not further specified.

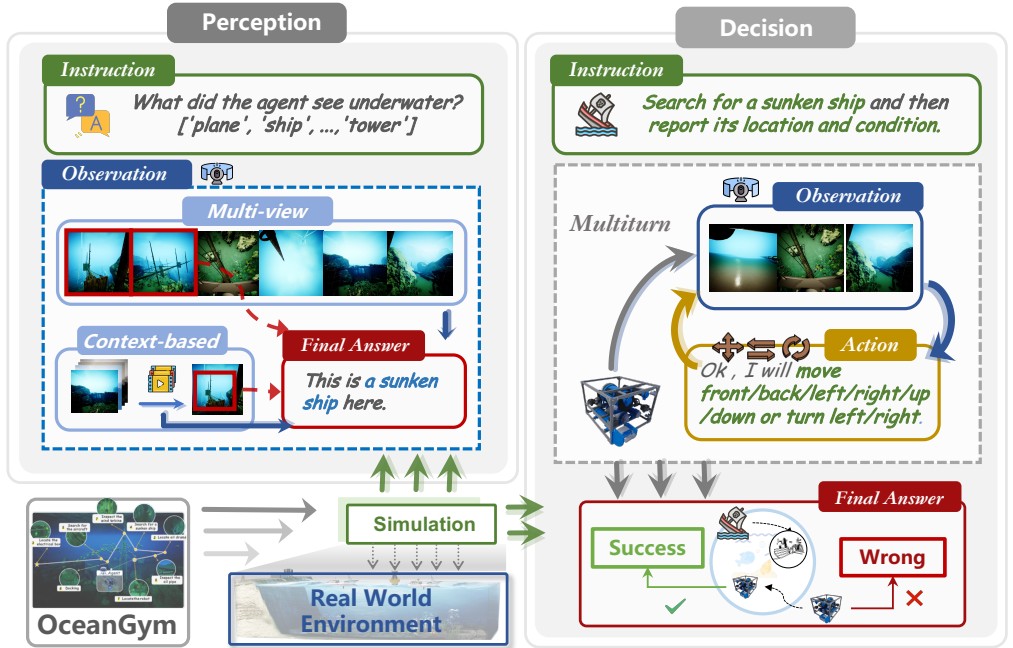

Figure 2: **OCEANGYM Tasks.** OCEANGYM comprises **Perception Tasks** (divided into **Multi-view Perception** and **Context-based Perception** settings) and **Decision Tasks** for evaluating embodied agents.

## 2.1 OCEANGYM ENVIRONMENT

We develop OCEANGYM atop Unreal Engine (UE) 5.3 (Epic Games, 2025), providing a comprehensive set of underwater environments, including both natural terrains and engineered structures. The environment features several semantic regions such as open water, seabed plains, underwater cliffs, pipeline networks, wreckage sites, and energy infrastructure zones. Each region is modeled with realistic physical and geometric properties, incorporating elements like oil pipelines, chemical waste barrels, submerged shipwrecks, electrical equipment, wind turbine foundations, and aircraft debris (more details in Appendix A.4). These elements are built using intricate 3D assets based on real-world references, ensuring accurate representation of structural and material characteristics.

We also simulate different lighting conditions by controlling the depth of the underwater environment. In our experiments, we configure two representative depths to emulate shallow (50m) and deep water (500m)[2] scenarios. For each task, the starting position is randomly selected to vary task difficulty, because tasks become increasingly challenging when the start point is far from the target, the target is initially out of view, or the initial orientation faces away from the goal. Furthermore, OCEANGYM is completely scalable, allowing users to customize the environment by selecting new depths to simulate more complex lighting conditions, or by adding new props and designing additional tasks based on the existing environment, thereby extending the diversity and difficulty.

## 2.2 UNDERWATER EMBODIED AGENTS

We model the agent's control–perception loop as a Partially Observable Markov Decision Process (POMDP) enhanced with contextual memory. At each time step $t$, the agent processes the task specification $\mathcal{T} = (I_{\text{target}}, c)$, where $I_{\text{target}}$ is a visual reference image of the target and $c$ provides its textual identity and features. It also considers language instruction $L$, synchronized observations $O_t$, and its memory state $m_t$. These elements collectively shape the agent's perception and objectives.

With the above information, the agent must generate either a textual perception response $y_t$ for perception tasks, or determine a control action $a_t$ for decision tasks. Here, $a_t \in \mathcal{A}$ is a discrete action selected from the action space $\mathcal{A}$. A decision trajectory is described by $\sigma = (O_1, a_1, s_1, m_1, \ldots, m_{t-1}, O_t, a_t, s_t)$. In this sequence, $O_i$ represents the observations, $a_i$ the actions, $s_i$ the states, and $m_i$ the memory states at each time step $i$. The episode concludes when the target is achieved or when the maximum time $t_{\max}$ is exhausted. The ultimate reward is based on the successful score of the task, as defined in §2.5.

---

[2]For deep water scenarios, optical sensing relies on artificial light sources, with a visibility range of approximately 0–10m.

**State and Observation.** The agent's state at time $t$ is given by $s_t = \{(x_t, y_t, z_t), (\phi_t, \theta_t, \psi_t)\}$, where $(x_t, y_t, z_t)$ represent the agent's 3D positional coordinates, and $(\phi_t, \theta_t, \psi_t)$ denote the roll, pitch, and yaw angles, respectively. At each timestep, the agent receives synchronized RGB and sonar images from sensors oriented in six different directions. The directions are defined by the set $\mathcal{D}_{\text{sensor}} = \{f, b, l, r, u, d\}$, corresponding to `front`, `back`, `left`, `right`, `up`, and `down`. The RGB images from these directions are denoted as $O_t^R = \{o_{t,d}^R\}_{d \in \mathcal{D}_{\text{sensor}}}$, and the sonar images are represented similarly as $O_t^S = \{o_{t,d}^S\}_{d \in \mathcal{D}_{\text{sensor}}}$. Therefore, the complete observation at time $t$ can be expressed as a combination of both image sets, $O_t = (O_t^R, O_t^S)$.

**Action Space.** The agent's action direction set is defined as $\mathcal{D}_{\text{action}} = \{f, b, l, r, u, d, rl, rr\}$, which encompasses both directional and rotational movements. Directional actions include translations along the primary axes: `forward` ($f$), `backward` ($b$), `left` ($l$), `right` ($r$), `up` ($u$), and `down` ($d$). Rotational actions consist of `rotate left` (rl) and `rotate right` (rr). At each timestep $t$, the agent selects an action $a_t \in \mathcal{A}$ from this discrete set and applies a control magnitude $\delta \in \mathbb{R}_{\geq 0}$ to determine the execution intensity.

**Memory.** Memory systems play a crucial role in storing and structuring past information, thereby enhancing the agent's resilience in dynamic and partially observable environments (Xi et al., 2025; Liu et al., 2023; Zhong et al., 2024; Wu et al., 2024; Maharana et al., 2024). OCEANGYM agent maintains an explicit memory $m_t$, structured as a sliding window that records the last $K$ steps:

$$m_t = \{(d_{t-k}, a_{t-k}) \mid k = 1, 2, \ldots, K\}. \tag{1}$$

Within this memory structure, $d_{t-k}$ denotes the textual description at time $t-k$, and $a_{t-k}$ represents the corresponding action executed. The sliding window size $K$ is implemented primarily to prevent the context length from exceeding the model's maximum input capacity. The default window size is large enough to capture the necessary historical information for most tasks in our benchmark. The perception module $\mathcal{P}_\theta$, modeled as an MLLM, generates a summary based on the current context and the interaction history $\{(O_k, a_k)\}_{k=t-K}^t$:

$$d_t = \mathcal{P}_\theta \big( \{(O_k, a_k)\}_{k=t-K}^t \big). \tag{2}$$

This summary is subsequently used to refresh the memory: $m_{t+1} = \text{update}(m_t, d_t, a_t)$.

**Memory-augmented Markov Process.** To maintain the Markov property while incorporating memory, we introduce an augmented hidden state $\tilde{s}_t = (s_t, m_t)$. The state transition is then modeled as:

$$p(\tilde{s}_{t+1} \mid \tilde{s}_t, a_t, \delta), \tag{3}$$

where $p(\cdot \mid \cdot)$ represents the augmented state transition function of the environment. This function captures both the evolution of memory, ensuring that the system remains Markovian despite the added complexity of memory integration.

**Agent Policy.** The agent policy is a multimodal, memory-augmented mapping parameterized by an MLLM with parameter vector $\theta$:

$$\pi_\theta(a_t, y_t \mid L, O_t, m_t, \mathcal{T}, \delta), \tag{4}$$

Concretely, for perception tasks, we sample an answer $y_t \sim \pi_\theta(y \mid L, O_t, m_t, \mathcal{T}, \delta)$, and for decision tasks, we sample an action $a_t \sim \pi_\theta(a \mid L, O_t, m_t, \mathcal{T}, \delta)$. An episode terminates at time $T$ when the agent either outputs a STOP command (for decision tasks) or provides a final answer to the question (for perception tasks) or when the maximum time $t_{\max}$ is reached. The policy, combined with the memory-augmented transition dynamics, induces the trajectory distribution:

$$\mathbb{P}_\theta(\sigma \mid L, \mathcal{T}) = \prod_{t=1}^{T-1} \pi_\theta(a_t, y_t \mid L, O_t, m_t, \mathcal{T}, \delta) \, p(\tilde{s}_{t+1} \mid \tilde{s}_t, a_t, \delta), \tag{5}$$

where $\sigma$ represents the trajectory of the agent through the state space over time, influenced by the specified policy $\pi_\theta$ and the transition model $p(\tilde{s}_{t+1} \mid \tilde{s}_t, a_t, \delta)$.

## 2.3 OCEANGYM PERCEPTION TASKS

The perception tasks are categorized into two settings: **Multi-View Perception** and **Context-based Perception**. These tasks primarily use RGB images as input, with sonar data added in certain experiments to enhance perception. The data for each setting are collected by human operators and designed to evaluate different aspects of MLLMs' perceptual abilities. There are a total of 85 scenes. More details in Appendix A.3.

**Multi-view Perception Setting.** This setting evaluates the agent's ability to interpret visual information from multiple synchronized viewpoints. At each timestep $t$, the agent captures a set of six simultaneous RGB images, denoted as $O_t^R = \{o_{t,d}^R\}_{d \in \mathcal{D}_{\text{sensor}}}$, where $d$ refers to the different sensor orientations: front, back, left, right, up, and down. The objective is to consistently identify and localize underwater objects across these varied viewpoints. This setting examines whether objects visible from certain angles can be correctly perceived when the visual inputs from all directions are sequentially processed by the MLLM, thereby evaluating robustness to viewpoint variations.

**Context-based Perception Setting.** This setting assesses the agent's ability to perceive and interpret sequential observations gathered during navigation. At each timestep $t$, the agent captures an RGB image $o_t^R$ from a fixed orientation, forming a chronological sequence $O_{1:m}^R = \{o_t^R\}_{t=1}^m$, where $m$ is the total number of timesteps. The agent must track and understand changes over time, ensuring consistent and accurate identification and localization of underwater objects. This evaluation emphasizes temporal consistency and the agent's capacity to build a coherent perception from a stable visual perspective in dynamic and complex underwater environments.

> **Running Example: Shipwreck Area**
>
> **Perception Task:** (1) Multi-view perception setting. The agent receives perception images (visual and sonar) from different sensors at the same time to determine the target, such as whether it is a shipwreck. (2) Context-based perception setting. The agent analyzes images one by one along a trajectory from a fixed viewpoint to identify the target.
>
> ---
>
> **Decision Task:** The agent receives a task instruction, such as "Search for a sunken ship," and then explores the area for 30 minutes to complete it.

## 2.4 OCEANGYM DECISION TASKS

**Decision Task Definition.** Decision tasks evaluate decision-making in continuous 3D environments, where agents must integrate multimodal sensory input with task specifications. Each episode begins from an initial state $s_0 = \{(x_0, y_0, z_0), (\phi_0, \theta_0, \psi_0)\}$ and requires the agent to reach the target defined by $\mathcal{T}$. The agent must combine sensory observations $O_t$, temporal memory, and goal information to execute precise maneuvers in cluttered, low-visibility environments. Key parameters of the task include the decision interval $t_{\text{interval}}$ and the task's limited duration $t_{\text{max}}$[3]. The decision interval $t_{\text{interval}}$ determines how frequently the agent makes decisions and executes actions. The total task duration $t_{\text{max}}$ sets the temporal constraint, within which the agent must meet its objectives, thereby influencing the planning and movement strategies employed by the agent. Compared with grid-based navigation benchmarks, this task emphasizes continuous control and realistic underwater environment, reflecting the challenges of autonomous exploration and inspection tasks.

**Decision Task Design.** To evaluate the decision-making capabilities of MLLMs in marine environments, we design eight representative task scenarios that are commonly used in actual underwater operations (more details in Appendix A.4). The task construction methods are divided into two categories: detection tasks and tracking tasks. Detection tasks focus on assessing the ability of MLLMs to locate specific underwater objects, including searching for large targets such as sunken ships or aircraft wreckage, and smaller targets like scientific research robots. Tracking tasks focus on evaluating the ability of MLLMs to perform inspection and monitoring tasks underwater, including scenarios like pipeline inspection and platform approaches. To further investigate the performance in challenging environments, four representative tasks are conducted under low light deep-sea conditions. In the experimental design, a systematic initial positioning strategy is adopted for each

---

[3]By default, $t_{\text{interval}}$ takes 30 seconds and $t_{\text{max}}$ takes 0.5 hours in decision tasks.

Table 1: Performance of perception tasks across different models and conditions. Values represent accuracy percentages (%). Adding sonar means using both RGB and sonar images.

| Model | Shallow Water Environment (High Illumination) | | | | | Deep Water Environment (Low Illumination) | | | | |
| | Multi-View Perception | | Context-based Perception | | Avg | Multi-View Perception | | Context-based Perception | | Avg |
| | Vision | +Sonar | Vision | +Sonar | | Vision | +Sonar | Vision | +Sonar | |
|---|---|---|---|---|---|---|---|---|---|---|
| GLM-4.5V | 52.73 | **56.36** | 46.67 | 63.33 | 54.77 | **36.36** | **30.91** | 20.00 | **33.33** | **30.15** |
| GPT-4o-mini | 34.55 | 34.55 | 20.00 | 33.33 | 30.61 | 14.55 | 20.00 | 3.33 | 6.67 | 11.14 |
| Gemini-2.5-Flash | 29.09 | 30.91 | 50.00 | 33.33 | 35.83 | 9.09 | 5.45 | 20.00 | 30.00 | 16.14 |
| Qwen2.5-VL-7B | **58.18** | 43.64 | **56.67** | **70.00** | **57.12** | 27.27 | 20.00 | 33.33 | **33.33** | 28.48 |
| Minicpm-4.5 | 52.73 | 43.64 | 36.67 | 23.33 | 39.09 | **29.09** | 23.64 | **43.33** | 13.33 | 27.35 |
| Human | 100.00 | 100.00 | 100.00 | 100.00 | 100.00 | 94.55 | 98.18 | 86.67 | 90.00 | 92.35 |

task. The first two starting positions remain consistent across all tasks to ensure experimental reproducibility. The third starting position is randomly generated within the operational boundary to evaluate the adaptability of the agent to different initial conditions.

## 2.5 EVALUATION METRICS

**Perception Task Evaluation.** We evaluate model performance using exact match accuracy. Let $y_i$ denote the ground-truth answer and $\hat{y}_i$ represent the model's predicted answer for the $i$-th sample.

$$\text{Acc} = \frac{100\%}{N} \sum_{i=1}^{N} \mathbb{I}\left[\hat{y}_i = y_i\right], \tag{6}$$

For multiple-choice items, $y_i$ and $\hat{y}_i$ are treated as sets and equality requires an exact set match.

**Decision Task Evaluation.** We evaluate decision tasks using a distance-based scoring method. Each episode ends when the agent issues a STOP command or reaches the time limit $t_{\max}$. For a task with $n$ evaluation points, let $\mathbf{p}_i$ be the $i$-th target location. If the target is detected, we use the closest position from the agent's trajectory to $\mathbf{p}_i$; otherwise, we use the agent's final position. The Euclidean distance is computed as $d_i = \|\hat{\mathbf{p}}_i - \mathbf{p}_i\|_2$, and the score for each point is defined as:

$$S_i = \begin{cases} 100, & d_i \leq \tau_1, \\ 100\,\dfrac{\tau_2 - d_i}{\tau_2 - \tau_1}, & \tau_1 < d_i \leq \tau_2, \\ 0, & d_i > \tau_2, \end{cases} \tag{7}$$

where the distance thresholds are set to $\tau_1 = 30$ meters and $\tau_2 = 100$ meters by default. The total score is a weighted sum as $S_{\text{total}} = \sum_{i=1}^{n} w_i S_i$, where $w_i$ are task-specific weights[4].

## 3 EXPERIMENTS

### 3.1 EXPERIMENTAL SETTINGS

To thoroughly evaluate the perception and decision capabilities of MLLMs in underwater environments, we conduct experiments using a variety of models[5]. Among the open-source models, we assess MiniCPM-V-4.5 (Yao et al., 2025), GLM-4.5V (Team et al., 2025) and Qwen2.5-VL-7B (Bai et al., 2025). For proprietary models, we test GPT-4o-mini (OpenAI, 2024) and Gemini-2.5-Flash (Gemini Team, 2024). **We run each task three times and report the average results.** Humans provide perception and decision-making answers based on tasks, and operate underwater robots through keyboards for decision-making tasks.

---

[4]For a single-point task $w_1 = 1.0$; for two points $(w_1, w_2) = (0.6, 0.4)$; for three points $(w_1, w_2, w_3) = (0.6, 0.2, 0.2)$.

[5]Note that our setup is designed to real-world deployment of MLLMs in the future; accordingly, we prioritize smaller-scale models that can run natively on edge devices.

Table 2: Performance in decision tasks requiring autonomous completion by MLLM-driven agents.

| Task | Model | | | | Human |
|---|---|---|---|---|---|
| | GLM-4.5V | GPT-4o-mini | Gemini-2.5 | Qwen2.5-VL-7B | |
| **Shallow Water Environment (High Illumination)** | | | | | |
| Locate the robot | $6.6_{\pm 19.83}$ | $\mathbf{8.9}_{\pm 10.1}$ | $0.0_{\pm 0.00}$ | $\underline{7.8}_{\pm 13.5}$ | 100 |
| Locate the oil drums | $\underline{10.7}_{\pm 16.52}$ | $\mathbf{11.1}_{\pm 19.2}$ | $3.5_{\pm 6.0}$ | $5.7_{\pm 9.8}$ | 100 |
| Locate the electrical box | $7.9_{\pm 17.08}$ | $\mathbf{36.6}_{\pm 21.9}$ | $\underline{15.9}_{\pm 27.4}$ | $8.7_{\pm 15.0}$ | 100 |
| Search for a sunken ship | $5.9_{\pm 5.04}$ | $\underline{13.4}_{\pm 19.3}$ | $\mathbf{20.5}_{\pm 14.3}$ | $10.3_{\pm 10.3}$ | 100 |
| Search for the aircraft | $\mathbf{25.0}_{\pm 6.10}$ | $\underline{16.9}_{\pm 17.8}$ | $11.7_{\pm 15.6}$ | $7.8_{\pm 10.0}$ | 100 |
| Inspect oil pipe | $\mathbf{37.8}_{\pm 17.88}$ | $27.1_{\pm 23.6}$ | $18.3_{\pm 15.8}$ | $\underline{30.8}_{\pm 25.2}$ | 100 |
| Inspect the wind turbine | $\underline{20.3}_{\pm 28.89}$ | $13.9_{\pm 14.33}$ | $\mathbf{25.1}_{\pm 22.1}$ | $14.7_{\pm 17.0}$ | 100 |
| Docking | $14.9_{\pm 13.20}$ | $\underline{19.2}_{\pm 33.28}$ | $\mathbf{19.4}_{\pm 33.6}$ | $8.3_{\pm 7.2}$ | 100 |
| **Average** | $\underline{16.1}_{\pm 15.6}$ | $\mathbf{18.4}_{\pm 19.9}$ | $14.4_{\pm 16.1}$ | $11.8_{\pm 13.7}$ | 100 |
| **Deep Water Environment (Low Illumination)** | | | | | |
| Locate oil drums | $\mathbf{10.6}_{\pm 21.35}$ | $\underline{5.6}_{\pm 9.69}$ | $0.0_{\pm 0.0}$ | $0.0_{\pm 0.0}$ | 40.8 |
| Search for a sunken ship | $2.9_{\pm 2.16}$ | $\mathbf{12.8}_{\pm 14.48}$ | $\underline{8.2}_{\pm 14.1}$ | $3.4_{\pm 5.8}$ | 100 |
| Inspect the oil pipe | $\mathbf{32.5}_{\pm 5.86}$ | $15.8_{\pm 15.5}$ | $6.6_{\pm 11.4}$ | $\underline{21.7}_{\pm 25.3}$ | 78.2 |
| Inspect the wind turbine | $0.0_{\pm 0.0}$ | $\mathbf{25.1}_{\pm 16.0}$ | $\underline{10.6}_{\pm 10.0}$ | $0.4_{\pm 0.6}$ | 100 |
| **Average** | $\underline{11.5}_{\pm 7.3}$ | $\mathbf{14.8}_{\pm 13.9}$ | $6.4_{\pm 8.8}$ | $6.4_{\pm 8.4}$ | 69.6 |

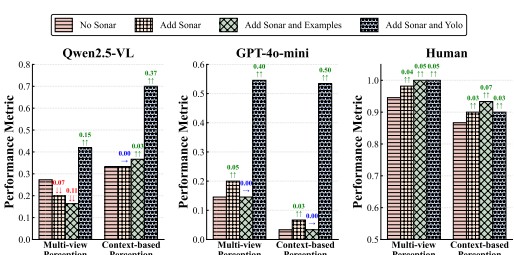

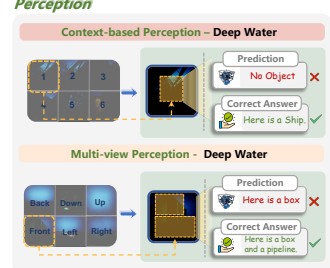

Figure 3: Performance comparison between human and MLLMs after adding sonar and sonar reference examples for objects in deep water environments.

Figure 4: Case analysis in perception tasks. Agents are susceptible to perception errors under challenging conditions such as low-light environments, multi-object scenarios, and occlusions.

## 3.2 MAIN RESULTS

**Perception Results.** The results for perception tasks are summarized in Table 1. In shallow, well-illuminated water environments, Qwen2.5-VL-7B achieves the strongest overall performance among the evaluated MLLMs, with an average accuracy of 57.12%, while GLM-4.5V demonstrates competitive performance with 54.77% average accuracy. Multi-view perception generally yields higher accuracy than the context-based setting across most models, likely because targets of similar size across viewpoints are easier to interpret, whereas distant objects in sequential views can introduce ambiguity. Under deep water conditions with low illumination, all models exhibit significant performance degradation, though GLM-4.5V emerges as the most robust (30.15% average accuracy), followed by Qwen2.5-VL-7B (28.48%) and Minicpm-4.5 (27.35%). Notably, incorporating sonar data does not consistently improve performance across models or tasks (further analysis in §3.3).

**Decision Results.** Performance on decision tasks is shown in Table 2. Several tasks resulted in zero scores, indicating extreme difficulty due to small object size or time constraints. GPT-4o-mini achieves the best average performance in both shallow (18.4%) and deep water (14.8%) environments, with GLM-4.5V ranking second under shallow conditions (16.1%) and deep water conditions (11.5%). Performance declines markedly in deep water, where Gemini-2.5 and Qwen2.5-VL-7B both average 6.4%. Notably, GLM-4.5V demonstrates strong performance in specific tasks, achieving the highest scores in "Search for the aircraft" (25.0%) and "Inspect oil pipe" (37.8%) in shallow water, and "Inspect the oil pipe" (32.5%) in deep water. Human performance substantially outperforms all models, reaching 100% in shallow water and 69.6% in deep water, underscoring the gap between current MLLM-driven decision-making and human proficiency.

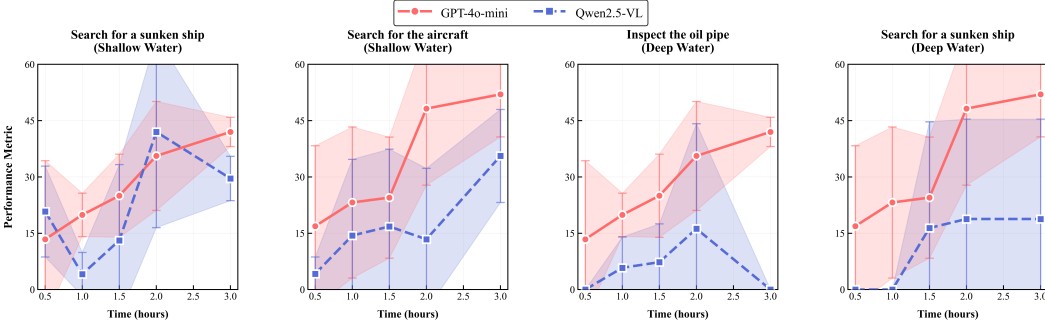

Figure 5: Scaling analysis performance over time in decision tasks.

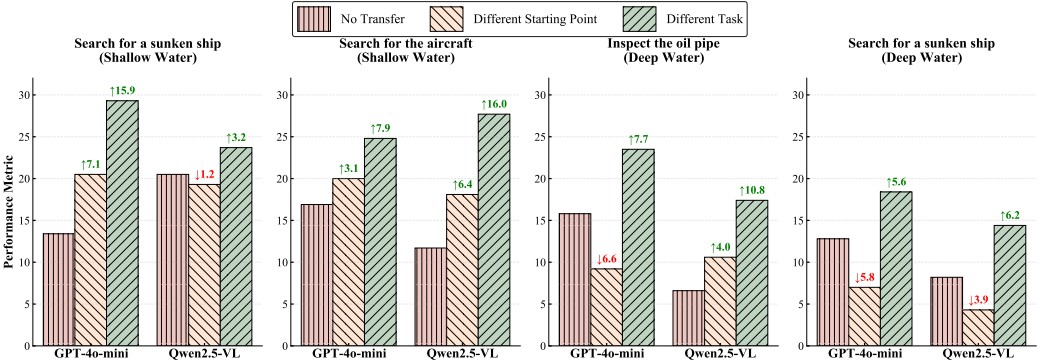

Figure 6: Impact of different memory transfer mechanisms on model performance.

## 3.3 ANALYSIS

**MLLM agents struggle to exploit sonar data for enhanced underwater perception, in stark contrast to humans who leverage it effectively.** To investigate the role of sonar data in deep-water environments, we compare the performance of human experts with the two MLLMs, Qwen2.5-VL and GPT-4o-mini, on perception tasks. Specifically, we either let the models directly comprehend sonar images or provide them with human-annotated interpretations as prompts. As shown in Figure 3, human experts consistently benefit from incorporating sonar data across tasks. By contrast, MLLMs exhibit only limited gains when using raw sonar images, and this gap becomes even more pronounced when reference sonar images of each object are introduced. This limitation likely stems from current MLLMs' fundamental difficulty in interpreting sonar imagery and underwater perceptual data (Xie et al., 2022; Zheng et al., 2023; Xu et al., 2025; Aubard et al., 2025), combined with potential constraints in the sonar simulation within OceanGym, an issue we discuss in §3.3. Notably, when employing a YOLO model (Redmon et al., 2016) specifically trained on sonar data as auxiliary perception tools, we observe significant performance improvements, suggesting that specialized vision models may currently outperform general-purpose MLLMs in sonar data interpretation tasks.

**Extended exploration enhances an agent's acquisition of environmental knowledge and task performance, following a scaling law that eventually plateaus.** We analyze the relationship between navigation performance and operational duration using the representative MLLMs, across both shallow- and deep-water scenarios. The performance was evaluated over durations of 0.5, 1, 1.5, 2, and 3 hours. As shown in Figure 5, performance initially improves with longer operation time, consistent with prior studies on test-time scaling (Zhang et al., 2025a; Zhu et al., 2025), but eventually plateaus. This plateau reflects inherent limitations in perception, memory, and reasoning, as well as a lack of intrinsic curiosity to explore new regions. These findings underscore the need to improve both fundamental MLLM capabilities and agent strategies, such as enhanced memory and long-horizon planning, to break through performance ceilings in embodied environments.

**Memory transfer enables agents to leverage past experience to tackle new challenges.** We investigate whether knowledge and experience accumulated from previous tasks (Hou et al., 2024; Hu et al., 2024a; Tan et al., 2025; Tang et al., 2025) can enhance performance in new tasks. Specifically, we explore using agents' previously explored trajectories as experiential input. Experiments are conducted in both shallow water and deep water environments, evaluating two transfer conditions: within-task transfer (different starting points) and cross-task transfer (different but related tasks). As

Figure 7: Case analysis in decision tasks.

shown in Figure 6, memory transfer improves decision-making performance in shallow water environments under both transfer conditions. However, in the more challenging deep water environment, only cross-task transfer demonstrates stable performance improvements, while within-task transfer shows limited benefits. This suggests that more appropriate prior experiences provide more robust guidance under perceptually degraded conditions. Transfer learning helps compensate for perceptual limitations by providing informed priors about environmental structure and effective navigation strategies. These findings underscore the importance of developing adaptive memory retrieval mechanisms that can selectively leverage relevant past experiences to enhance decision-making in autonomous underwater agents operating under diverse environmental conditions.

**Case analysis.** We present case analyses and illustrate failure cases in Figure 4, mainly due to: **(1) Occlusions**, where targets are partially blocked; **(2) Multi-object Scenes**, causing identification and localization ambiguities; and **(3) Low Illumination**, which severely reduces vision-based perception accuracy. Figure 7 shows common decision task failures, primarily from: **(1) Perception Errors**, where inaccurate detection leads to wrong actions; and **(2) Memory Forgetting**, where the agent cannot retain crucial past information, such as visited locations or previous decisions. Furthermore, we deploy physical objects that serve as real-world references for object modeling in OceanGym into an actual marine environment to correlate simulated performance with real-world performance. An AUV equipped with a sonar data acquisition system is then used to collect sonar measurements. As shown in the Figure 8, **the YOLO model trained in the simulated environment enhances GPT-4o-mini's ability to interpret real-world sonar data**. However, it exhibits limited generalization capability for objects not included in the simulation.

**Discusses and Limitations of OCEANGYM.** OCEANGYM offers a versatile testbed for underwater embodied agents, though it cannot fully replicate real-world conditions as factors like currents, salinity, marine life, and sonar noise remain imperfectly modeled. Despite these constraints, OCEANGYM supports synthetic data generation and facilitates reinforcement learning with rich feedback, and serves as a sim-to-real bridge for deploying models on AUVs (See §A.2).

## 4 RELATED WORK

**Embodied Simulations.** Embodied intelligence describes artificial intelligence systems whose intelligent behavior emerges through continuous physical and sensory interactions with the environment (Gupta et al., 2021; Ding et al., 2024; Shi et al., 2024). Simulation platforms are essential for advancing such systems across ground, aerial, and marine domains (Liu et al., 2024b; Han et al., 2025; Aldhaheri et al., 2025). In ground applications, platforms like Matterport3D (Chang et al., 2017), House3D (Wu et al., 2018), and Habitat (Manolis Savva et al., 2019) provide realistic indoor and outdoor environments for navigation, scene understanding, and human-robot interaction research. Aerial robotics benefits from simulators such as AirSim (Shah et al., 2018), CityNav (Lee et al., 2025) and OpenUAV (Wang et al., 2024a), which offer high-fidelity simulations with accurate physics and sensor models. Similarly, in the marine domain, simulation platforms such as HoloOcean (Potokar et al., 2022), OceanSim (Song et al., 2025), and MarineGym (Chu et al., 2025) provide specialized capabilities for modeling hydrodynamic effects and underwater dynamics. With the development of embodied intelligence, an increasing variety of simulation environments (Kolve

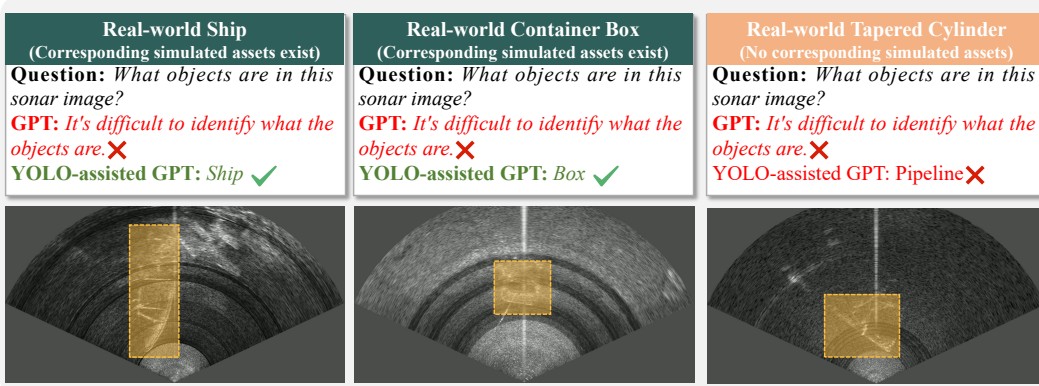

Figure 8: We evaluate whether YOLO models trained in simulated environments can enhance real-world performance by testing them on actual sonar data. The results demonstrate that while the YOLO-assisted GPT-4o-mini approach yields measurable performance improvements for certain objects modeled in OceanGym, the models exhibit limited generalization capability for objects not included in the simulation.

et al., 2017; Puig et al., 2018; Xiang et al., 2020; Gan et al., 2021; Li et al., 2021; Nasiriany et al., 2024; Zhou et al., 2024b; Hong et al., 2025) have emerged to meet specific tasks, needs, or scenarios.

**MLLM-driven Embodied Agents.** Building upon the rapid advancement of LLMs (Achiam et al., 2023; Touvron et al., 2023; Chiang et al., 2023; Yang et al., 2025a), the emergence of MLLMs (OpenAI, 2024; Bai et al., 2025; Meta AI, 2024; Liu et al., 2024a; Gemini Team, 2024; Team et al., 2025; Wang et al., 2025b) has further strengthened agent capabilities by incorporating visual understanding for multimodal perception. Despite impressive results in various agent applications (Hu et al., 2024b; Ning et al., 2025), MLLM-driven agents still face substantial challenges in real-world and simulated embodied environments. Key difficulties persist in spatial cognition (Prasad et al., 2023; Du et al., 2024; Tong et al., 2024; Shiri et al., 2024; Zheng et al., 2024; Dang et al., 2025; Yang et al., 2025c; Li et al., 2025), task planning (Chen et al., 2023; Huang et al., 2023; Zhou et al., 2024a) , object navigation (Wang et al., 2024b; Khanna et al., 2024; Guo et al., 2025; Qiao et al., 2025; Cheng et al., 2025), and robotic manipulation (Zheng et al., 2022a; Yang et al., 2025b; Wang et al., 2025a). To evaluate agent capabilities, embodied benchmarks have been developed across diverse settings, including indoor (Anderson et al., 2018; Wu et al., 2018), urban (Chen et al., 2019; Caesar et al., 2020; Vasudevan et al., 2021; Gao et al., 2024), aerial (Yao et al., 2024; Gao et al., 2025b; Cai et al., 2025), specialised (Zheng et al., 2022b; Luo et al., 2023; Song et al., 2024; Li et al., 2024a) and real-world (Zhao et al., 2025; Koh et al., 2024; Zhang et al., 2025b) scenarios.

## 5 CONCLUSION

We introduce OCEANGYM, the first bechmark environment specifically designed for underwater embodied agents. Our experiments reveal significant limitations in current MLLMs. We hope OCEANGYM can bridge the gap between simulated research and real-world deployment, offering a foundation for developing robust autonomous systems for marine applications.

ETHICS STATEMENT

This research is conducted in strict compliance with established ethical guidelines and best practices in scientific research. All data employed in this study are obtained from publicly accessible datasets, with no utilization of proprietary or confidential information. Proper and accurate citations are provided for all data sources referenced throughout this paper. We emphatically advise all users to maintain the highest ethical standards when utilizing our dataset, ensuring principles of fairness, transparency, and responsibility in their research applications. Any use of the dataset that may potentially cause harm or adversely affect societal welfare is expressly prohibited.

REPRODUCIBILITY STATEMENT

We provide data from our benchmark under file size limitation, along with the corresponding evaluation code, in the supplementary materials. Detailed descriptions of the environment setup and data construction procedures are available in § 2.1, § 2.3 and § 2.4. Additional data details and

comprehensive benchmark statistics can be found in Appendix A.3 and Appendix A.4. Specific configurations of the tested models are documented in Section 3.1.

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

# A   APPENDIX

## A.1   THE USE OF LARGE LANGUAGE MODELS (LLMs)

We confirm that LLMs are used only as an auxiliary tool to assist in refining wording and sentence structure. Their application in experiments is strictly confined to scientific research purposes, and all such uses have been clearly documented in the Experimental Settings. No additional reliance on LLMs has been involved in this work.

## A.2   MORE DETAILED DISCUSSES AND LIMITATIONS

**Limitations**. While OCEANGYM provides a valuable testbed for underwater embodied agents, several limitations should be acknowledged. First, OceanGym leverages Unreal Engine (UE) 5.3 (Epic Games, 2025) for realistic underwater environment rendering and physical simulation, while utilizing HoloOcean's (Potokar et al., 2022) cluster-based multipath ray-tracing algorithm to simulate multibeam sonar. Although UE plugins can be used to simulate water flow, buoyancy, lighting, water interaction etc, it cannot fully replicate the real underwater environment, as factors such as ocean currents, salinity, marine life, and geological changes are not accurately captured. Future work may leverage generative models (Ball et al., 2025) or physics-informed machine learning to incorporate these complexities. The optical and sonar images still differ from those in the real world, particularly since sonar simulation introduces errors. We will continue to refine the system to reduce these discrepancies, noting that real-world sonar itself is also subject to noise and inaccuracies. In addition, the environment is large and requires considerable computational resources, with at least 24GB of GPU memory. We recommend running without a graphical interface, as enabling it can cause significant lag. These limitations highlight opportunities for future work to expand task coverage, improve physical realism, and optimize computational efficiency.

**Applications of OceanGym.** (1) A competitive arena for evaluating foundational models and embodied agent frameworks, particularly memory mechanisms. Future work can leverage OCEANGYM to optimize prompt design, memory utilization, and base model capabilities. (2) A platform for synthesizing underwater simulation data to enhance both perception and decision-making skills of agents. (3) A testbed for reinforcement learning, providing rich feedback for training autonomous behaviors. (4) A sim-to-real bridge, enabling the transfer of trained models to real-world AUVs. By connecting virtual training with real-world deployment, OCEANGYM substantially reduces dependence on costly and hazardous field trials, accelerates development cycles, and enhances the reliability and robustness of autonomous underwater systems.

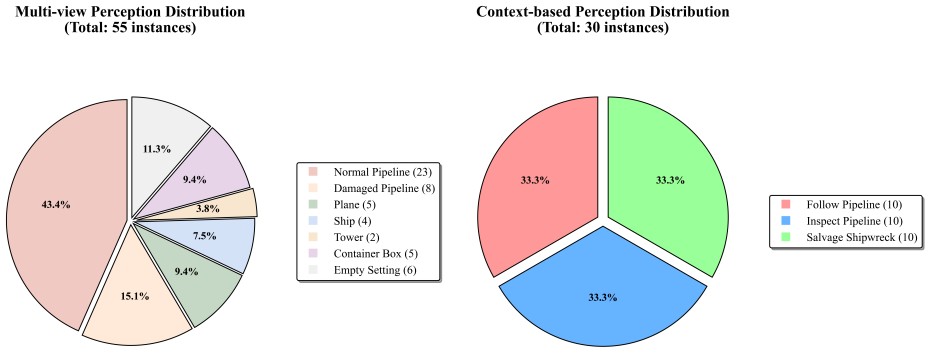

Figure 9: Statistics of perception tasks.

## A.3   PERCEPTION TASK STATISTICS

Figure 9 presents the statistical distribution of different perception settings analyzed in our dataset. The dataset consists of 85 sets of data, which include 55 sets focusing on Multi-view Perception and 30 sets on Context-based Perception. Within the Multi-view Perception data, 55 sets are categorized as follows: 23 sets involve normal pipelines, 8 sets entail damaged pipelines, 5 sets are related to

planes, 4 sets concern ships, 2 sets focus on towers, 5 sets involve container boxes, and 6 sets do not feature any specific dominant object. For the Context-based Perception data, the 30 sets are evenly divided among three distinct sub-tasks, each comprising 10 sets. These sub-tasks involve the agent following pipelines, inspecting pipelines for potential damage, and scanning around shipwrecks.

### A.4 DECISION TASK DETAILS

Decision-making tasks require an embodied agent to accomplish a given objective through a series of decisions. Figure 10 illustrates the perceptual input at one specific state during such a task.

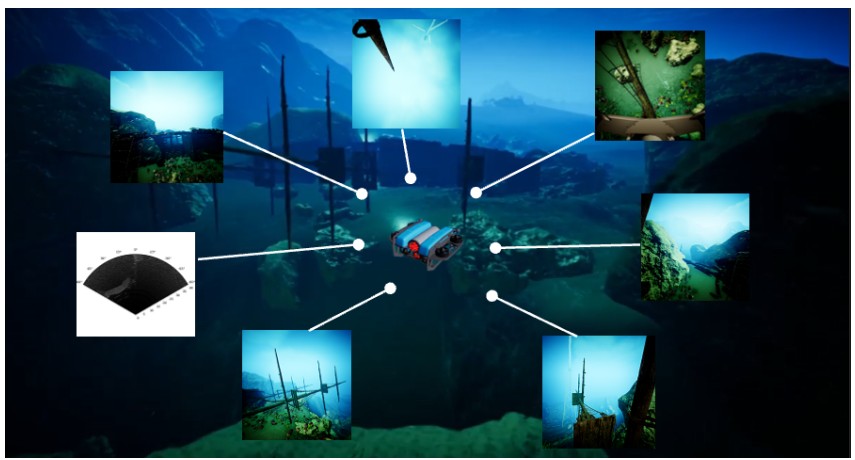

Figure 10: A state case of a decision-making task.

**Locate the robot.**   Locate and approach the mining robot in a complex underwater environment within an abandoned subsea research zone characterized by variable terrain, low visibility, and artificial structures. The operational protocol mandates an initial memory check for the target's coordinates; if available, the system engages in direct coordinate-based navigation. Absent prior data, the robot utilizes its six camera feeds for visual comparison against a reference image, identifying the target by its distinct shape, structure, and color. A systematic exploration pattern, such as a grid or linear search, is then executed. Throughout the mission, all encountered special objects and artificial structures are documented. Maintaining a strict minimum standoff distance of 10 meters from all rocks and obstacles is the highest priority, superseding all other actions. The vehicle must remain within the predefined operational boundaries at all times, and all reports must exclusively detail artificial structures, explicitly ignoring any marine life.

**Inspect the oil pipe.**   Locate and identify the abandoned subsea oil pipeline network situated in a central zone where pipelines may be partially buried and serve as potential navigation references. The procedure begins with a query of the robot's memory for known pipeline coordinates, initiating direct navigation if the data is present. Without prior coordinates, the robot employs its camera feeds to detect linear structures and surface features that match the reference imagery of a pipeline. This is followed by a systematic exploration of the area to comprehensively document all artificial structures and special objects. A critical safety requirement is to maintain a safe distance from all obstacles, executing immediate directional changes upon hazard detection. All reporting must focus solely on artificial structures, with biological entities entirely omitted from logs.

**Locate oil drumss.**   Locate and identify oil drums or barrels submerged in an environment where they may be partially buried or scattered within sediment under conditions of poor visibility. The first action is a memory scan for stored coordinates of oil drums, proceeding with direct waypoint navigation if the search is successful. If no coordinates exist, the robot must use its camera systems to identify cylindrical objects and any visible markings that align with the target description. A methodical search pattern is then conducted across the operational area, with all special objects documented. Strict obstacle avoidance protocols are continuously enforced, and the robot's trajectory must never exceed the designated operational boundaries. Reports are confined to artificial structures and special objects only.

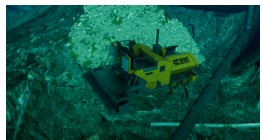
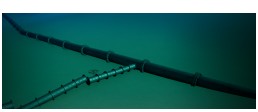
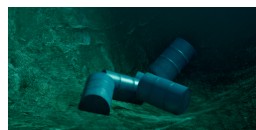
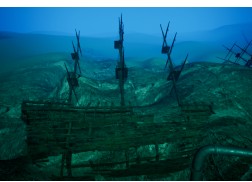

Figure 11: Target object for the "Locate the robot" task.

Figure 12: Target object for the "Inspect the oil pipe" task.

Figure 13: Target object for the "Locate oil drums" task.

Figure 14: Target object for the "Search for a sunken ship" task.

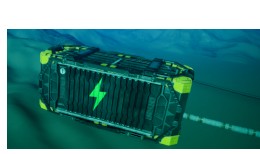
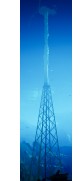
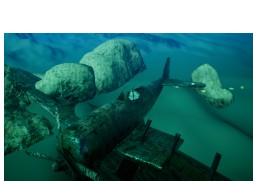
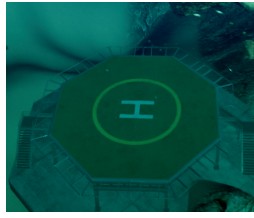

Figure 15: Target object for the "Locate the electrical box" task.

Figure 16: Target object for the "Inspect the wind turbine" task.

Figure 17: Target object for the "Search for the aircraft" task.

Figure 18: Target object for the "Docking" task.

**Search for a sunken ship.** Locate and identify sunken shipwrecks, which are typically structurally complex entities that may be partially buried or obscured by various underwater obstacles. The mission initiates with an access of the robot's memory for any known coordinates of shipwrecks, utilizing them for direct navigation if available. In the absence of positional data, the robot relies on its camera feeds to recognize large structural features and surface details that correspond to the reference images of a shipwreck. A systematic exploration is subsequently performed to document all special objects within the area. A safe distance from all obstacles must be maintained throughout the operation, and the vehicle is required to stay within its prescribed operational limits. All marine life is systematically ignored and excluded from reporting.

**Locate the electrical box.** Locate and identify underwater electrical boxes, which are often partially buried in sediment and possess distinctive structural features. The operational sequence starts with a retrieval attempt from the robot's memory for the coordinates of electrical boxes, followed by direct navigation to any located waypoints. Without prior coordinate data, the robot must analyze its camera feeds to identify the target based on its specific shape, structural characteristics, and any identifiable markings. A thorough and systematic exploration of the zone is then carried out, with all special objects recorded. The mission must adhere to strict obstacle avoidance procedures and remain within the defined operational boundaries at all times. All communications and reports are restricted to artificial structures and special objects.

**Inspect the wind turbine.** Locate and identify underwater wind power station structures, which are large installations featuring multiple pillars and mechanical components. The robot first searches its internal memory for stored coordinates of the wind power station, navigating directly to the location if the data is found. If the coordinates are not located, the system uses its camera arrays to identify the major structural and mechanical elements that match the reference documentation. A systematic exploration pattern is executed to document every special object in the vicinity. A safe buffer distance from all obstacles is perpetually maintained, and the robot's path must comply strictly with the operational boundaries. Any biological entities encountered are disregarded and not included in any reports.

**Search for the aircraft.** Locate and identify underwater aircraft wreckage, which can be complex and potentially dispersed across different areas of the seafloor. The initial phase involves a memory check for any existing coordinates related to aircraft wreckage, with immediate navigation initiated upon a successful find. If no data is available, the robot switches to using its visual feeds to identify key structural features and surface details that are consistent with the target wreckage. A comprehensive systematic search is then conducted, ensuring all special objects are documented. Strict obstacle avoidance is paramount, and the vehicle must operate entirely within the set boundaries. Reports are exclusively to contain information on artificial structures and special objects.

**Docking.** Locate and identify an underwater landing platform marked with a distinctive "H" symbol, a structure with a regular form that provides a reliable navigation reference. The robot's first action is to consult its memory for the platform's coordinates, proceeding with direct navigation if the information is available. Should the coordinates be absent, the platform must be identified visually through the camera feeds by recognizing the "H" marking and the overall platform structure. This is followed by a systematic exploration to document all special objects in the area. A safe distance from all obstacles must be maintained, and the operation is confined to the approved boundaries. All reporting is limited to artificial structures and special objects, with no mention of biological activity.

## A.5    PROMPT FOR OCEANGYM

---

**Prompt for Perception Tasks**

**[RGB Image]**
You are an assistant that analyzes an image and checks which of the following options appear in it.
Options:**[Options]**
Instructions:
- Carefully examine the image, even the corners.
- You can choose single or multiple options, if none of the options appear, just return an empty list.
- For multiple-choice questions, no points will be awarded for incomplete selections, over-selections, or incorrect selections.
- The output must be a valid list (only list, no explanation, no extra text).

---

**Prompt for Perception Tasks (Add Sonar)**

**[Sonar Image]**
This sonar image can be used as a reference to assist in identifying the next color image.

**[RGB Image]**
You are an assistant that analyzes an image and checks which of the following options appear in it. Before that, I have already provide you a sonar image to help you choose the correct one.
Options:**[Options]**
Instructions:
- Only when you find it difficult to recognize the color image, I suggest you refer to the previous sonar image together.
- Carefully examine the image, even the corners.
- You can choose single or multiple options, if none of the options appear, just return an empty list.
- For multiple-choice questions, no points will be awarded for incomplete selections, over-selections, or incorrect selections.
- The output must be a valid list (only list, no explanation, no extra text).

---

**Prompt for Perception Tasks (Add Sonar and Examples)**

**[Object A Sonar Image]**
This sonar image example is **[Object A]**.

**[Object B Sonar Image]**
This sonar image example is **[Object B]**.

**...**

**[Sonar Image]**

---

This sonar image can be used as a reference to assist in identifying the next color image.

**[RGB Image]**
You are an assistant that analyzes an image and checks which of the following options appear in it. Before that, I have already provide you a sonar image to help you choose the correct one.
Options:[Options]
Instructions:
- Only when you find it difficult to recognize the color image, I suggest you refer to the previous sonar image together.
- Carefully examine the image, even the corners.
- You can choose single or multiple options, if none of the options appear, just return an empty list.
- For multiple-choice questions, no points will be awarded for incomplete selections, over-selections, or incorrect selections.
- The output must be a valid list (only list, no explanation, no extra text).

---

Prompt for Navigation Tasks

You are an expert pilot for an Autonomous Underwater Vehicle (AUV), designated as the "Control Expert". Your mission is to navigate a complex underwater environment to complete specific tasks. You will receive data from six cameras and location sensors. Your decisions must be precise, safe, and strategic.
1. Tactical Briefing for the Area of Operations
Before the mission begins, you must internalize the following intelligence about the operational area. This context is vital for interpreting sensor data and forming a macro-level strategy.

...

3. Mission Briefing and Sensor Data
Task Description: [Task Description]
Target Object Name: [Object Name]
Target Object Reference Image: [Object Image]
Target Object Description: [Object Description]

...

5. Survey Navigation Commands
Available Commands: 'ascend', 'descend', 'move left', 'move right', 'move forward', 'move backward', 'rotate left', 'rotate right', 'stop'.
Command Execution: You must only issue ONE command per turn from the list above.

...

Remember:
Conduct comprehensive reconnaissance! Systematic coverage = priority! Use efficient exploration patterns! Catalog all special objects! Maintain exploration momentum! Always use format! Ignore all marine life! One continuous line between markers!

Table 3: Performance of perception tasks across different prompts.

| Model | Shallow Water Environment (High Illumination) | | | | | Deep Water Environment (Low Illumination) | | | | |
|---|---|---|---|---|---|---|---|---|---|---|
| | Multi-View Perception | | Context-based Perception | | Avg | Multi-View Perception | | Context-based Perception | | Avg |
| | Vision | +Sonar | Vision | +Sonar | | Vision | +Sonar | Vision | +Sonar | |
| GPT-4o-mini(prompt1) | 34.55 | 34.55 | 20.00 | 33.33 | 30.61 | 14.55 | 20.00 | 3.33 | 6.67 | 11.14 |
| GPT-4o-mini(prompt2) | 54.55 | 45.45 | 40.00 | 30.00 | 42.5 | 20.00 | 20.00 | 10.00 | 0.00 | 12.5 |

## A.6 THE IMPACT OF DIFFERENT PROMPTS ON PERCEPTION TASKS.

Due to the difficulty in finding a prompt that is suitable for all MLLMs, we test the impact of different prompts on the model. As shown in Table 3, we find that the impact was relatively small in deep water environment. Prompt1 is the prompt used in the main experiment, and prompt2 is the best prompt for GPT-4o-mini during the testing process.

