# OpenReview forum: "OceanGym: A Benchmark Environment for Underwater Embodied Agents"
_ICLR.cc/2026/Conference — ICLR 2026 Conference Withdrawn Submission_

### Official Review · Reviewer_dEvG · 2025-10-20

**Soundness:** 2
**Presentation:** 3
**Contribution:** 2
**Rating:** 4
**Confidence:** 4

**Summary:**

This paper presents **OceanGym**, a simulation benchmark for evaluating **multimodal large language model (MLLM) agents** in underwater environments. Built on Unreal Engine and HoloOcean, it integrates RGB and sonar sensing with discrete movement-based actions. The agent operates under a **memory-augmented Markov process**, where a fixed-size sliding window stores recent textual summaries and actions generated by an MLLM. The benchmark includes **85 test scenarios** spanning perception and decision-making tasks, such as object localization, inspection, and navigation under shallow and deep-water conditions. Quantitative results compare different MLLM backbones on perception accuracy and distance-based decision metrics. Overall, OceanGym provides a unified platform for studying embodied multimodal reasoning and language-driven control in simulated underwater settings, combining perception, memory, and decision-making within a single evaluation framework.

**Strengths:**

1. The paper addresses an important and timely problem—applying AI and multimodal reasoning to underwater robotics and autonomous exploration, which has significant real-world value.
2. The manuscript is clearly structured and easy to follow. The motivation, design choices, and experimental setup are described in a coherent and readable way.
3. The authors have implemented a complex simulation and benchmarking platform that integrates Unreal Engine, multimodal sensors, and an MLLM-based agent framework. This represents a considerable amount of engineering work and system integration.

**Weaknesses:**

1. The environment and control formulation are overly simplified, which may limit the realism and potential sim-to-real applicability of OceanGym. In particular, the **discrete “movement-based” action space** (instead of physically grounded velocity or thrust control) and the **idealized environment dynamics** (absence of current, drag, or inertia) create a noticeable gap between simulation and real-world AUV behavior.
2. The paper designs the agent’s memory as a fixed-size sliding window. This setup inherently limits the agent’s ability to capture **long-horizon temporal dependencies**, since any information beyond the most recent K steps is forgotten. As a result, the analysis and claims regarding the agent’s memory or its ability to “remember” become theoretically weak because of this design weakness.
3. The benchmark includes only 85 test scenes, and the discrete percentages in Table 1 (e.g., 33.33 %) suggest that each condition was evaluated with only one or very few samples. This limited evaluation scope introduces high result variance and reduces the credibility of the reported averages. Consequently, the dataset lacks the diversity needed to claim robust or generalizable agent performance.

**Questions:**

1. See Weaknesses.
2. In the Memory Transfer experiment, the paper states that the agent leverages “knowledge and experience accumulated from previous tasks.” Could the authors clarify whether this experience is derived from **the agent’s own previously sampled trajectories** (correct or not?) or from **teacher/expert trajectories**? The current description suggests self-generated experience, but it is not clear.
3. Your memory mechanism is a fixed-size sliding window of textual summaries and past actions generated by an MLLM$((m_t={(d_{t-k},a_{t-k})}_{k=1}^K))$. Conceptually, this looks close to “detect objects → record what was seen → remember recent actions.”
**Have you evaluated a non-MLLM classical baseline** that uses (i) a conventional CV detector for RGB/sonar, plus (ii) a **rule-based summarizer** to produce the same kind of per-step text summary (object, pose/status, etc)? This pipeline may be more accurate and efficient since you do not leverage MLLM much. I found [1] has a similar implementation to my description.

[1]. Zhang, Hongxin, et al. "Building cooperative embodied agents modularly with large language models." *arXiv preprint arXiv:2307.02485* (2023).

---

> ### Author Response · Authors · 2025-11-21
> **Response to Reviewer dEvG [1/2]**
>
> Dear Reviewer dEvG,
>
> **Thank you for your constructive feedback.** We appreciate your recognition of our work's strengths: **timely research problem with real-world significance**, **clear and coherent presentation**, and **substantial technical implementation of an integrated simulation platform**.
> Below, we respond to each concern in turn.
>
> ### **Response to Weakness:**
>
> > For W1: The simplified environment and discrete action space in OceanGym create a significant gap between simulation and real-world autonomous underwater vehicle operations.
>
> The **discrete action space follows established embodied AI conventions for evaluating core capabilities**, while the **platform's extensible architecture supports integration of complex physics** like velocity control and fluid dynamics.
> - We acknowledge the importance of velocity and thrust control in real operations. However, our current use of a discrete action space **follows established practices in many terrestrial and aerial embodied environments and datasets**[1][2], which successfully employ **similar abstractions to evaluate core model capabilities**.
> - OceanGym supports **modifications to buoyancy, propulsion parameters, and object placement**. The platform also maintains compatibility with **Unreal Engine's fluid plugins for simulating water current interactions**. These features provide viable pathways to incorporate more complex physical dynamics in future iterations.
>
> > For W2: The agent's fixed-size sliding window memory limits its ability to capture long-term dependencies, weakening the theoretical foundation for claims about its memory capacity.
>
> We apologize for any confusion caused by our initial description. The fixed-size sliding window memory is implemented primarily to **prevent the context length from exceeding the model's maximum input capacity**. The **window size is set very large**, which proves sufficient to capture necessary historical information for most tasks in our benchmark without truncating critical dependencies.
>
> > For W3: The benchmark's limited scale and lack of diversity, with only 85 test scenes and very few samples per condition, introduce high result variance and undermine claims of robust agent performance.
>
> Our dataset is **carefully designed and collected by humans to ensure diversity**, incorporating multiple perspectives and starting positions for each object. While the current evaluation scope uses 85 scenes, we observe **consistent model performance across multiple samplings of the same prompt**, with small result variance.
> We have added more model results. And experiments (```Table 3```) have been conducted to investigate **the effects of different prompts, which have been found to be controllable, especially for difficult problems**. The result is as follows:
>
> | Model                        | Shallow Water Environment (High Illumination) | Deep Water Environment (Low Illumination) |
> |------------------------------| --- | --- |
> | GPT-4o-mini (default prompt) | 30.61 | 11.14 |
> | GPT-4o-mini (best prompt)    | 42.5 | 12.5 |
>
> [1] OpenFly: A Versatile Toolchain and Large-scale Benchmark for Aerial Vision-Language Navigation
>
> [2] Vision-and-Dialog Navigation

---

> ### Author Response · Authors · 2025-11-21
> **Response to Reviewer dEvG [2/2]**
>
> ### **Response to Questions:**
>
> > For Q1: See Weaknesses.
>
> See weaknesses response.
>
> > For Q2: The paper needs to clarify whether the "accumulated experience" in memory transfer comes from the agent's own trajectories or from expert demonstrations.
>
> Sorry for the unclear parts. The experience in the memory transfer experiment is derived from the **agent's own previously sampled trajectories**.
>
> > For Q3: A suggested alternative is to use a classical pipeline with a conventional computer vision detector and a rule-based summarizer instead of an MLLM, as it may be more accurate and efficient.
>
> Thanks for your suggestion. We have evaluated a complementary approach (```Fig. 3 and Sec. 3.3, Para 1```)  where **a YOLO model trained on sonar data serves as an auxiliary perception module**. Providing its predictions to the MLLM improves the model's utilization of sonar information. The results are as follows:
>
>
> | Model          | Shallow Water - Multi-View (+Sonar) | Shallow Water - Context-based (+Sonar) | Deep Water - Multi-View (+Sonar) | Deep Water - Context-based (+Sonar) |
> |----------------|------------------------------------:|---------------------------------------:|---------------------------------:|------------------------------------:|
> | GPT-4o-mini    |                              34.55% |                                 33.33% |                           20.00% |                               6.67% |
> | GPT-4o-mini (+YOLO) |                          **54.55%** |                             **63.33%** |                       **54.55%** |                          **53.33%** |
> | Qwen2.5-VL-7B  |                              43.64% |                                 70.00% |                           20.00% |                              33.33% |
> | Qwen2.5-VL-7B (+YOLO) |                          **52.73%** |                                 70.00% |                       **41.82%** |                          **70.00%** |
>
> ---
>
> Thanks again for your time and precious advice!
>
> We hope our response can address your concerns. **Please let us know if you have any further questions**.
>
> Best Regards,
>
> Authors of OceanGym.

---

> ### Author Response · Authors · 2025-11-27
> **Kindly Reminder**
>
> Dear Reviewer dEvG,
>
> We sincerely thank you for your constructive feedback and valuable suggestions on our work. With the discussion period ending in a few days, we kindly encourage you to share any further questions or insights to allow sufficient time for our response. ﻿
>
> If our revisions have addressed some of your concerns, we would greatly appreciate it if you could consider raising your rating for our paper. ﻿
>
> Thank you very much for your time and consideration. We look forward to your reply.
>
> Best Regards,
>
> Authors of OceanGym.

---

### Official Review · Reviewer_usJW · 2025-10-31

**Soundness:** 1
**Presentation:** 2
**Contribution:** 1
**Rating:** 2
**Confidence:** 4

**Summary:**

The paper introduces OceanGym, a benchmark for underwater embodied agents, constructed upon UE. The paper introduces 8 tasks, including perception and decision-making tasks. The paper benchmarks 4 MLLMs and human performance, revealing a large gap.

**Strengths:**

- The paper is well-motivated, designing environments and benchmarks to study underwater embodied agents is important.

**Weaknesses:**

- The contribution of the proposed platform is questionable.
  - Underwater setting is challenging because of the dynamic ocean currents, salinity, ... but all these factors are not modeled.
  - How the sensor data is simulated is unclear.
  - How the proposed platform could align with real-world conditions is unclear, like what the real underwater robots is being referred to when designing the agents in the paper?
  - What's the contribution compared to previous work, like HoloOcean mentioned in the paper?

- Missing details for the benchmark.
  - There are no task statistics in the main paper.
  - There is no example of what the sensor data looks like, like the sonar images.
  - How human performance is evaluated is unclear.
  - Human performance on the benchmark is 100 and nearly 100 for perception tasks, and not reported for decision-making tasks.

- Missing related literature discussion
  - There is much more related literature in the embodied simulations and embodied agents domain, just naming a few. [1][2][3]
  - Appendix A.2 More Related Works is still empty.

[1] BEHAVIOR-1K: A Human-Centered, Embodied AI Benchmark with 1,000 Everyday Activities and Realistic Simulation
[2] ThreeDWorld: A Platform for Interactive Multi-Modal Physical Simulation
[3] HAZARD Challenge: Embodied Decision Making in Dynamically Changing Environments

**Questions:**

See Weaknesses.

---

> ### Author Response · Authors · 2025-11-21
> **Response to Reviewer usJW**
>
> Dear Reviewer usJW,
>
> **Thank you for your constructive feedback.** We appreciate your recognition of our work's strengths: **well-motivated research direction** and **the importance of studying underwater embodied agents**.
> Below, we respond to each concern in turn.
>
> ### **Response to Weakness:**
>
> > For W1: The platform's contribution is questioned due to its omission of key real-world challenges, unclear sensor simulation, lack of alignment with actual underwater robots, and undefined advantages over existing simulators.
>
> OceanGym's contribution is validated through its **extensible real-world physics simulation**, **explicit alignment with actual AUV specifications**, and **integration of embodied AI frameworks for underwater scenarios**. We have conducted a supplement (```Appx. A.6 and Fig 9```).
> - OceanGym supports **modifications to buoyancy, propulsion, and object placement**, and is compatible with **Unreal Engine's fluid plugins for simulating water current interactions**. These capabilities provide pathways to incorporate various environmental factors.
> - **RGB sensor data is generated using Unreal Engine**, while **sonar data is simulated using cluster-based multipath ray-tracing algorithms based on HoloOcean**.
> - The platform's design references **real-world Autonomous Underwater Vehicles (AUVs)** (```Sec. 1, Para Building OceanGym and Fig 1```). By **aligning the simulation’s sensor inputs, action spaces, and control protocols with those used in real AUVs**, OceanGym serves as a functional sim-to-real bridge, reducing the need for costly field trials to some extent. OceanGym is compatible with Robot Operating System (ROS) commands, which allows seamless integration with real-world AUVs. This design enables agents tested or trained in our simulated environment to be directly deployed on physical platforms for decision-making, control, and communication tasks.
> - Although there are some underwater simulation environments, **they do not bridge simulation platforms with embodied AI agent frameworks**. In addition, we **design tasks and evaluation that reflect ocean scenarios**, and provided **our analysis and insights through experiments**. Comparative analysis with HoloOcean:
>
> | Feature                  | OceanGym                                      | HoloOcean                                         |
> |:-------------------------|:----------------------------------------------|:--------------------------------------------------|
> | Environment              | Large environment with various task scenarios | A small scene with very few objects          |
> | Embodied Agent Framework | MLLM-driven agent framework                   | Not provided                                      |
> | Task Design              | Based on realistic underwater tasks           | Not provided                                      |
> | Task Evaluation          | Comprehensive metrics provided                | Not provided                                      |
>
> > For W2: The benchmark lacks essential details, including task statistics, examples of sensor data, and a clear explanation of how human performance was evaluated.
>
> The missing benchmark details have been supplemented.
> - **Detailed task statistics** are provided in ```Appendix. A.3```. We have **added some task statistics** in the revised draft (```Sec. 2.3```) for better visibility.-
> - **Figure 1** includes **examples of simulated sensor data**. The revised draft also incorporates **additional sonar image examples** (```Appx. A.6 and Fig 9```) for further clarification.
> - **Human performance is evaluated by individuals unfamiliar with the environment**. They complete **perception tasks by providing answers** and **decision-making tasks by operating the robot via keyboard controls**. Their performance is then scored using the evaluation metrics defined in the paper.
> - **Human performance is strong in well-lit shallow water conditions but declines noticeably in deep water environments.** The results for **human performance on decision-making tasks are already provided** (```Table 2```).
>
> > For W3: Missing related literature discussion.
>
> We have **expanded the related literature section to include the suggested references and other relevant works**. The revised draft now contains a more detailed related works. (```Sec. 4```).
>
>
> ---
>
> Thanks again for your time and precious advice!
>
> We hope our response can address your concerns. **Please let us know if you have any further questions**.
>
> Best Regards,
>
> Authors of OceanGym.

---

> ### Author Response · Authors · 2025-11-27
> **Kindly Reminder**
>
> Dear Reviewer usJW,
>
> We sincerely thank you for your constructive feedback and valuable suggestions on our work. With the discussion period ending in a few days, we kindly encourage you to share any further questions or insights to allow sufficient time for our response. ﻿
>
> If our revisions have addressed some of your concerns, we would greatly appreciate it if you could consider raising your rating for our paper. ﻿
>
> Thank you very much for your time and consideration. We look forward to your reply.
>
> Best Regards,
>
> Authors of OceanGym.

---

### Official Review · Reviewer_C1Qr · 2025-11-01

**Soundness:** 3
**Presentation:** 3
**Contribution:** 4
**Rating:** 4
**Confidence:** 4

**Summary:**

The paper introduces a new embodied environment, the OceanGym, that focuses on underwater scenarios where perception and planning are all difficult. The paper then tests the environment using MLLM agents that are given only the current agent state and the perception result.

**Strengths:**

1. The proposed challenge has very useful practical applications as well as adequate difficulty for the community to explore.

2. The tasks and metrics are all well-formed and can be adapted easily by other embodied planning methods, making development easy on the environment.

**Weaknesses:**

1. MLLM performing poorly can be anticipated. However, the paper lacks evidence that the poor performance comes from the difficulty of underwater setting instead of internal weakness of MLLMs themselves. Maybe MLLM just can't handle the planning even not in underwater environments.

2. Results show that adding sonar on top of RGB makes MLLM perform worse. This suggests potentially ill-designed prompts of MLLM baselines.

3. The environment only simulates the lighting, not other extreme conditions underwater, which could mean large deployment gap.

4. There could be more in-domain vision-based methods tested on the environment.

**Questions:**

1. Difficulties underwater include not only perception, but also low-level controlling / disturbance, e.g. due to water currents. Can your environment simulate this in some way?

2. Have you tried training a vision module, e.g., that transforms a poorly lighted image to a well-lit one, or with depth prediction / bounding box prediction?

3. How do the baselines perform without the lighting constraints?

---

> ### Author Response · Authors · 2025-11-21
> **Response to Reviewer C1Qr [1/2]**
>
> Dear Reviewer C1Qr,
>
> **Thank you for your constructive feedback.** We appreciate your recognition of our work's strengths: **practical applicability with appropriate challenge level**, and **well-designed tasks and metrics with good adaptability**.
> Below, we respond to each concern in turn.
>
> ### **Response to Weakness:**
>
> > For W1: The paper fails to prove that the poor performance is due to the challenging underwater setting rather than the inherent planning limitations of MLLMs themselves.
>
> We acknowledge that MLLMs face inherent challenges in complex planning tasks. However, recent embodied benchmarks[1][2] demonstrate that **MLLMs can achieve reasonable performance in navigation and decision-making within terrestrial and aerial environments**, particularly with specialized methods[3][4] to enhance their capabilities. The significantly poorer performance of these MLLMs in OceanGym and previous works[5] suggests that the underwater setting presents unique and additional challenges beyond general MLLM limitations. In addtion, previous research[5] and our own tests on real-world marine data[6][7], indicates that **current MLLMs generally struggle to interpret sonar imagery and other specialized underwater optical data effectively**.  This challenge is also reflected in the OceanGym results.
>
> > For W2: The performance drop when adding sonar to RGB data suggests the MLLM baselines may have been used with poorly designed prompts.
>
> Our experiments reveal that current MLLMs struggle significantly with sonar interpretation. We also input cases from real sonar data[6][7] into MLLMs, but the response was still unsatisfactory. For GPT-4o-mini, **we tested multiple prompts and selected the optimal one, yet incorporating sonar data still showed no consistent improvement** (```Table 3```). The results are shown in the table below:
>
>
> | Model                 | Shallow Water (High Illumination) | | Deep Water (Low Illumination) | |
> |:---------------------|:---------------------------------:|:-:|:-----------------------------:|:-:|
> |                       | Multi-View +Sonar | Context-based +Sonar | Multi-View +Sonar | Context-based +Sonar |
> | GPT-4o-mini (prompt1) | 34.55             | 33.33                | 20.00             | 6.67                 |
> | GPT-4o-mini (prompt2) | 45.45             | 30.00                | 20.00             | 0.00                 |
>
> > For W3: The environment only simulates the lighting, not other extreme conditions underwater, which could mean large deployment gap.
>
> While OceanGym experiments currently focus on lighting variations, the **platform supports modifications to buoyancy, moving force, and object placement.** Furthermore, it is compatible with **fluid plugins to simulate water current interactions (e.g., fluid-like flow)**. These features provide pathways to incorporate additional environmental factors.
>
> > For W4: There could be more in-domain vision-based methods tested on the environment.
>
> We have added more experiments (```Fig. 3 and Sec. 3.3, Para 1```) in the revised draft. Our experiments using **a YOLO model trained on sonar data as an auxiliary tool have shown better results**. Furthermore, OceanGym is designed to support **rapid and convenient testing of various vision-based approaches**. The results are shown in the table below:
>
>
>
>
> | Model          | Shallow Water - Multi-View (+Sonar) | Shallow Water - Context-based (+Sonar) | Deep Water - Multi-View (+Sonar) | Deep Water - Context-based (+Sonar) |
> |----------------|------------------------------------:|---------------------------------------:|---------------------------------:|------------------------------------:|
> | GPT-4o-mini    |                              34.55% |                                 33.33% |                           20.00% |                               6.67% |
> | GPT-4o-mini (+YOLO) |                          **54.55%** |                             **63.33%** |                       **54.55%** |                          **53.33%** |
> | Qwen2.5-VL-7B  |                              43.64% |                                 70.00% |                           20.00% |                              33.33% |
> | Qwen2.5-VL-7B (+YOLO) |                          **52.73%** |                                 70.00% |                       **41.82%** |                          **70.00%** |
>
>
> [1] EmbodiedBench: Comprehensive Benchmarking Multi-modal Large Language Models for Vision-Driven Embodied Agents
>
> [2] Static and Plugged: Make Embodied Evaluation Simple
>
> [3] CityNavAgent: Aerial Vision-and-Language Navigation with Hierarchical Semantic Planning and Global Memory
>
> [4] FlightGPT: Towards Generalizable and Interpretable UAV Vision-and-Language Navigation with Vision-Language Models
>
> [5] NAUTILUS: A Large Multimodal Model for Underwater Scene Understanding
>
> [6] A Dataset with Multibeam Forward-Looking Sonar for Underwater Object Detection
>
> [7] Open-set recognition with long-tail sonar images

---

> ### Author Response · Authors · 2025-11-21
> **Response to Reviewer C1Qr [2/2]**
>
> ### **Response to Questions:**
>
> > For Q1: Difficulties underwater include not only perception, but also low-level controlling/disturbance, e.g. due to water currents. Can your environment simulate this in some way?
>
>
> Yes, OceanGym supports **modifications to buoyancy, moving force, and object placement**. Furthermore, it is compatible with **Unreal Engine's fluid plugins to simulate water current interactions**. These capabilities provide pathways to incorporate such environmental factors.
>
> > For Q2: Have you tried training a vision module, e.g., that transforms a poorly lighted image to a well-lit one, or with depth prediction/bounding box prediction?
>
> We conducted experiments using **a YOLO model trained on sonar data as an auxiliary tool for bounding box prediction**, and the experimental results are shown in W4.
>
> > For Q3: How do the baselines perform without the lighting constraints?
>
> Under high illumination conditions, **baseline performance shows improvement compared to low illumination scenarios**. However, as detailed in our case studies (```Sec. 3.3, Para Case analysis```), **agents still face persistent challenges in both perception and decision-making tasks even under favorable lighting conditions**.
>
> ---
>
> Thanks again for your time and precious advice!
>
> We hope our response can address your concerns. **Please let us know if you have any further questions**.
>
> Best Regards,
>
> Authors of OceanGym.

---

> > ### Comment · Reviewer_C1Qr · 2025-11-27
> >
> > The author adequately settles my main concern about underwater settings. I will raise my score to weak accept but I expect to see more experimental results on the underwater setting in later versions.

---

> > > ### Author Response · Authors · 2025-11-28
> > > **Thank you for your support, and we will conduct more experiments.**
> > >
> > > Dear Reviewer C1Qr,
> > >
> > > We are sincerely grateful for your positive feedback and your decision to increase the score.  Meanwhile, thank you for your suggestion. We are currently conducting more experiments. We will reply to you with new experiments and results as soon as possible, and update the paper.
> > >
> > > Best Regards,
> > >
> > > Authors of OceanGym.

---

> > > > ### Author Response · Authors · 2025-12-01
> > > > **More experimental results on the underwater setting**
> > > >
> > > > Dear Reviewer C1Qr,
> > > >
> > > >  **We deploy physical objects that serve as real-world references for object modeling in OceanGym into an actual marine environment to correlate simulated performance with real-world performance**(```Sec. 3.3, Para Case analysis  and  Fig 8```). An AUV equipped with a sonar data acquisition system is then used to collect sonar measurements. **The YOLO model trained in the simulated environment enhances GPT-4o-mini's ability to interpret real-world sonar data**. However, **it exhibits limited generalization capability for objects not included in the simulation**.
> > > >
> > > >
> > > > Best Regards,
> > > >
> > > > Authors of OceanGym.

---

> ### Author Response · Authors · 2025-11-27
> **Kindly Reminder**
>
> Dear Reviewer C1Qr,
>
> We sincerely thank you for your constructive feedback and valuable suggestions on our work. With the discussion period ending in a few days, we kindly encourage you to share any further questions or insights to allow sufficient time for our response. ﻿
>
> If our revisions have addressed some of your concerns, we would greatly appreciate it if you could consider raising your rating for our paper. ﻿
>
> Thank you very much for your time and consideration. We look forward to your reply.
>
> Best Regards,
>
> Authors of OceanGym.

---

### Official Review · Reviewer_dKUd · 2025-11-01

**Soundness:** 2
**Presentation:** 3
**Contribution:** 2
**Rating:** 4
**Confidence:** 4

**Summary:**

The manuscript proposes a benchmark environment for underwater embodied agents. The manuscript claims that the proposed environment will provide a basis for developing agents that are robust to challenging perceptual settings (e.g., low-light and low-visibility conditions, forced reliance on unconventional state estimation signals such as sonar) and difficult planning/reasoning tasks in the underwater environment.

**Strengths:**

- The paper is well-written and well-organized
- The manuscript proposed a novel benchmark for a lesser studied subfield of embodied AI
- The manuscript provides a reasonable set of initial experiments that provide sufficient headroom for subsequent research

**Weaknesses:**

- L88-89: Whereas the ultimate goal is to provide a pathway for transferring learned skills to real-world underwater vehicles, I am missing sufficient justification that the real-world underwater robotics community can really benefit from this environment. Put a different way, the manuscript lacks experiments illustrating that agents developed to achieve high performance in the proposed benchmark will be successful in the real world settings.
- Table 1, Section 3.3p1: The addition of sonar information in most cases seems to decrease the model performance. This yields a potential paradox in leveraging the proposed benchmark environment for developing better agents (see comment above): is is worthwhile to develop better agents in the proposed benchmark? Would agents that model vision + sonar information in the benchmark more effectively also perform better in real world environments or on other benchmarks?

**Questions:**

- Section 3.3p1: Could the authors provide some more rationale about why the proposed benchmark may be well-suited for the ICLR community, specifically? I feel as if the fit would be stronger if the manuscript provided some methodological/algorithmic insights as well, e.g., by providing stronger baselines that are better-suited to processing these multisensory information (as opposed to taking foundation models off the shelf and displaying low performance)

---

> ### Author Response · Authors · 2025-11-21
> **Response to Reviewer dKUd [1/2]**
>
> Dear Reviewer dKUd,
>
> **Thank you for your constructive feedback.** We appreciate your recognition of our work's strengths: **clear organization**, **novel benchmark in an under-explored research area**, and **well-designed experiments with strong extensibility**.
> Below, we respond to each concern in turn.
>
> ### **Response to Weakness:**
>
> > For W1: The manuscript lacks real-world experiments to prove that skills learned in the simulation will benefit actual underwater robotics.
>
> OceanGym bridges simulation and reality by aligning with real AUV standards, enabling synthetic data generation and rapid algorithm testing to reduce real-world deployment costs.
> 1. By **aligning the simulation's sensor inputs, action spaces, and control protocols** with those used in real Autonomous Underwater Vehicles (AUVs), OceanGym serves as a functional sim-to-real bridge, reducing the need for costly field trials to some extent. OceanGym is compatible with Robot Operating System (ROS) commands, which allows seamless integration with real-world AUVs. This design enables agents tested or trained in our simulated environment to be directly deployed on physical platforms for decision-making, control, and communication tasks.
> 2. Previous work in embodied AI for terrestrial and aerial domains has shown that simulation environments can be effectively used to **generate synthetic data** for model training[1][2][3]. This synthetic data can help improve the perception and decision-making capabilities of agents, thereby supporting sim-to-real transfer. In addition, OceanGym can be used for **reinforcement learning guided by environmental feedback** and **iterative improvement of agent capabilities** through various algorithmic approaches.
> 3. OceanGym serves as a testbed for **rapid evaluation**. It enables researchers to efficiently assess the effectiveness of various algorithms or models within challenging marine environments before committing to costly and logistically complex real-world trials.
>
>
> > For W2: The value of the benchmark is questioned, as adding sonar，a key real-world sensor，paradoxically reduces performance in the simulation.
>
> The observed performance drop with sonar integration precisely reveals a critical real-world limitation of MLLMs, making the benchmark valuable
> Our analysis, supported by previous research[4][5] and our own tests on real-world marine data[6][7], indicates that **current MLLMs generally struggle to interpret sonar imagery and other specialized underwater optical data effectively**. This challenge is also reflected in the OceanGym results, where the addition of sonar data did not consistently lead to performance gains. It exposes a significant and realistic gap in MLLMs' capabilities. Their inability to effectively fuse heterogeneous sensor data like vision and sonar, which is a skill crucial for real-world underwater operations. Therefore, we believe that developing better agents is a worthwhile and necessary try.
>
>
> [1] UrbanVideo-Bench: Benchmarking Vision-Language Models on Embodied Intelligence with Video Data in Urban Spaces
>
> [2] FlightGPT: Towards Generalizable and Interpretable UAV Vision-and-Language Navigation with Vision-Language Models
>
> [3] EmbSpatial-Bench: Benchmarking Spatial Understanding for Embodied Tasks with Large Vision-Language Models
>
> [4] NAUTILUS: A Large Multimodal Model for Underwater Scene Understanding
>
> [5] Marinegpt: Unlocking secrets of ocean to the public
>
> [6] A Dataset with Multibeam Forward-Looking Sonar for Underwater Object Detection
>
> [7] Open-set recognition with long-tail sonar images

---

> ### Author Response · Authors · 2025-11-21
> **Response to Reviewer dKUd [2/2]**
>
> > For Q1:  The manuscript should better justify the benchmark's relevance to ICLR by providing methodological insights and stronger, tailored baselines for multisensory data, rather than just using low-performing models.
>
> Thanks for your suggestion. OceanGym addresses a critical gap by providing the first underwater embodied AI benchmark, and has incorporated specialized perception models as stronger baselines
> Compared to other scenarios, there is currently **a lack of underwater embodied environments for testing agents**. And, we design a series of tasks for underwater operations for evaluation.
> In response to the reviewer's suggestion to explore stronger, more suitable baselines, we have conducted experiments (```Fig. 3 and Sec. 3.3, Para 1```) integrating a specialized perception model as tool. Specifically, we fine-tune a YOLO model on sonar imagery for object detection and use its outputs to augment the MLLM-based agent. The results are as follows:
>
>
>
> | Model          | Shallow Water - Multi-View (+Sonar) | Shallow Water - Context-based (+Sonar) | Deep Water - Multi-View (+Sonar) | Deep Water - Context-based (+Sonar) |
> |----------------|------------------------------------:|---------------------------------------:|---------------------------------:|------------------------------------:|
> | GPT-4o-mini    |                              34.55% |                                 33.33% |                           20.00% |                               6.67% |
> | GPT-4o-mini (+YOLO) |                          **54.55%** |                             **63.33%** |                       **54.55%** |                          **53.33%** |
> | Qwen2.5-VL-7B  |                              43.64% |                                 70.00% |                           20.00% |                              33.33% |
> | Qwen2.5-VL-7B (+YOLO) |                          **52.73%** |                                 70.00% |                       **41.82%** |                          **70.00%** |
>
> ---
>
> Thanks again for your time and precious advice!
>
> We hope our response can address your concerns. **Please let us know if you have any further questions**.
>
> Best Regards,
>
> Authors of OceanGym.

---

> ### Author Response · Authors · 2025-11-27
> **Kindly Reminder**
>
> Dear Reviewer dKUd,
>
> We sincerely thank you for your constructive feedback and valuable suggestions on our work. With the discussion period ending in a few days, we kindly encourage you to share any further questions or insights to allow sufficient time for our response.
> ﻿
>
> If our revisions have addressed some of your concerns, we would greatly appreciate it if you could consider raising your rating for our paper.
> ﻿
>
> Thank you very much for your time and consideration. We look forward to your reply.
>
> Best Regards,
>
> Authors of OceanGym.

---

> > ### Comment · Reviewer_dKUd · 2025-11-27
> >
> > I appreciate the authors’ response and effort in satisfying my concerns.
> >
> > The authors clarified AUV alignment and sensor modeling, but real-world validation remains conceptual rather than empirically demonstrated. It would be better if the manuscript could provide some experimental results, e.g., correlating performance in the proposed simulated benchmark with the referenced real-world marine data in [6] [7].
> >
> > The added augmented baselines and expanded motivation improve ICLR relevance, though the contribution remains primarily an environment rather than a methodological advance.
> >
> > I am willing to increase my score, but I would really just like to see some experiments that attempt to correlate simulated performance with real-world performance, if just on the basis of any limited sensor traces that are available.

---

> > > ### Author Response · Authors · 2025-11-28
> > > **Thank you for your reply and suggestions, and we will conduct relevant experiments.**
> > >
> > > Dear Reviewer dKUd,
> > >
> > > Thank you for your constructive suggestion and response. We are currently designing and conducting some experiments to correlate simulated performance with real-world performance, and will reply to you with new experiments and results as soon as possible.
> > >
> > > Best Regards,
> > >
> > > Authors of OceanGym.

---

> > > > ### Author Response · Authors · 2025-12-01
> > > > **More experiments that correlate simulated performance with real-world performance**
> > > >
> > > > Dear Reviewer dKUd,
> > > >
> > > >  **We deploy physical objects that serve as real-world references for object modeling in OceanGym into an actual marine environment to correlate simulated performance with real-world performance**(```Sec. 3.3, Para Case analysis  and  Fig 8```). An AUV equipped with a sonar data acquisition system is then used to collect sonar measurements. **The YOLO model trained in the simulated environment enhances GPT-4o-mini's ability to interpret real-world sonar data**. However, **it exhibits limited generalization capability for objects not included in the simulation**.
> > > >
> > > >
> > > > Best Regards,
> > > >
> > > > Authors of OceanGym.

---

### Author Response · Authors · 2025-11-21
**Summary of Advantages and Motivation [1/2]**

Thanks for your dedication and thoughtful reviews.

### Summary of Advantages

**First, we appreciate all of your positive comments highlighting the strengths of our work for a summary:**

- **Well-Motivated and Timely Contribution**
- "The manuscript proposed a novel benchmark for a lesser studied subfield of embodied AI." (Reviewer dKUd)
- "The paper is well-motivated, designing environments and benchmarks to study underwater embodied agents is important." (Reviewer usJW)
- "The paper addresses an important and timely problem—applying AI and multimodal reasoning to underwater robotics and autonomous exploration, which has significant real-world value." (Reviewer dEvG)

- **Clear Practical Value and Adaptability**
- "The proposed challenge has very useful practical applications as well as adequate difficulty for the community to explore." (Reviewer C1Qr)
- "The tasks and metrics are all well-formed and can be adapted easily by other embodied planning methods, making development easy on the environment." (Reviewer C1Qr)

- **Solid Experimental Foundation**
- "The manuscript provides a reasonable set of initial experiments that provide sufficient headroom for subsequent research." (Reviewer dKUd)

- **High-Quality Technical Execution**
- "The authors have implemented a complex simulation and benchmarking platform that integrates Unreal Engine, multimodal sensors, and an MLLM-based agent framework. This represents a considerable amount of engineering work and system integration." (Reviewer dEvG)

- **Excellent Presentation and Clarity**
- "The paper is well-written and well-organized." (Reviewer dKUd)
- "The manuscript is clearly structured and easy to follow. The motivation, design choices, and experimental setup are described in a coherent and readable way." (Reviewer dEvG)


### Our Motivation and Contributions

**We wish to reiterate the motivation and main contributions of our work.**

While several underwater simulation environments exist, none have been integrated with a comprehensive embodied agent framework tailored for intelligent decision-making and evaluation. This gap is particularly critical given the unique challenges posed by underwater settings, including extreme perceptual conditions and complex multi-modal sensing requirements.

This paper introduces OceanGym, the first comprehensive benchmark environment specifically designed for underwater embodied agents, bridging simulation platforms with embodied AI agent frameworks.

**Concretely, OceanGym provides:**
- A **configurable underwater simulation environment** supporting dynamic adjustment of multiple environmental parameters.
- **Eight realistic task domains** covering search, inspection, and docking scenarios;
- A **unified MLLM-driven agent framework** integrating perception, memory, and decision-making;
- **Comprehensive evaluation metrics** for both perception and decision tasks.
- Using **real sonar data to correlate simulated performance with real-world performance**.

Built on OceanGym, we conduct **extensive experiments** with state-of-the-art MLLMs, revealing significant performance gaps between current models and human experts. **Our analysis provides valuable insights** into the limitations of current MLLMs in processing sonar data, maintaining long-term consistency, and adapting to dynamic underwater conditions. **We will release OceanGym as an open platform** to support future research in robust underwater embodied intelligence.

Best Regards,

Authors of OceanGym.

---

> ### Author Response · Authors · 2025-11-21
> **Our Main Revisions [2/2]**
>
> We sincerely appreciate the constructive feedback and questions from the reviewers to improve our manuscript. In response to the concerns and questions raised by the reviewers, we have made the following revisions (our revised manuscript has been uploaded and added content marked in blue.):
>
> **For Reviewer dKUd:**
>
> - Add more analysis on how to benefit from this environment. (```Page 2, Sec. 1, Line 84-88```)
> - Add more experiments, related works and analysis on sonar utilization. (```Page 7, Fig. 3 and Page 8, Sec. 3.3, Para 1```)
> - Add more detailed related works. (```Page 9-10, Sec. 4```)
> - Add more experiments to correlate simulated performance with real-world performance. (```Page 9, Sec. 3.3, Para Case analysis  and  Page 10, Fig 8```)
>
> **For Reviewer C1Qr:**
>
> - Add more experiments, related works and analysis on sonar utilization. (```Page 7, Fig. 3 and Page 8, Sec. 3.3, Para 1```)
> - Add more detailed related work. (```Page 9-10, Sec. 4```)
> - Add more information about simulating water flow, sonar, etc. (```Page 19, Appx. A.2, Para 1```)
> - Add more experiments on the impact of different prompts. (```Page 23-24, Appx. A.6 and Tab 3```)
> - Add more experiments to correlate simulated performance with real-world performance. (```Page 9, Sec. 3.3, Para Case analysis  and  Page 10, Fig 8```)
>
> **For Reviewer usJW:**
>
> - Add more analysis on how to benefit from this environment. (```Page 2, Sec. 1, Line 84-88```)
> - Add more details to human evaluation. (```Page 6, Sec. 3.1```)
> - Add more experiments, related works and analysis on sonar utilization. (```Page 7, Fig. 3 and Page 8, Sec. 3.3, Para 1```)
> - Add more detailed related works. (```Page 9-10, Sec. 4```)
> - Added more description and presentation of our environment. (```Page 20, Appx. A.6 and Fig 10```)
>
> **For Reviewer dEvG:**
>
> - Add more details about the memory window. (```Page 4, Sec. 2.2, Line 186-187```)
> - Add more model results on our benchmark. (```Page 6-7, Sec. 3.2 Para 1 - Para 2 and Tab 1-Tab 2```)
> - Add more experiments, related works and analysis on sonar utilization. (```Page 7, Fig. 3 and Page 8, Sec. 3.3, Para 1```)
> - Add more detailed related works. (```Page 9-10, Sec. 4```)
> - Add more experiments on the impact of different prompts. (```Page 23-24, Appx. A.6 and Tab 3```)
>
> **We thank the reviewers for their thoughtful comments and suggestions**. We have revised the manuscript to address the raised points and believe these improvements help better present our work. The feedback has been valuable in strengthening the paper's clarity and technical soundness. We are happy to provide further clarification if needed. **We will continue to improve the quality and content of this draft based on discussions with all reviewers.**
>
>
> Best Regards,
>
> Authors of OceanGym.

---

### Note · Authors · 2025-12-25

I have read and agree with the venue's withdrawal policy on behalf of myself and my co-authors.